# Cross-talk between QseBC and PmrAB two-component systems is crucial for regulation of motility and colistin resistance in Enteropathogenic *Escherichia coli*

**Blanca Fernandez-Ciruelos**[1]*, **Tasneemah Potmis**[1], **Vitalii Solomin**[2], **Jerry M. Wells**[1]*

**1** Host-Microbe Interactomics Group, Wageningen University & Research (WUR), Wageningen, The Netherlands, **2** Organic Synthesis Methodology Group, Latvian Institute of Organic Synthesis (LIOS), Riga, Latvia

* blanca.fernandezciruelos@wur.nl (BF-C); jerry.wells@wur.nl (JMW)

## Abstract

The quorum sensing two-component system (TCS) QseBC has been linked to virulence, motility and metabolism regulation in multiple Gram-negative pathogens, including Enterohaemorrhagic *Escherichia coli* (EHEC), Uropathogenic *E. coli* (UPEC) and *Salmonella enterica*. In EHEC, the sensor histidine kinase (HK) QseC detects the quorum sensing signalling molecule AI-3 and also acts as an adrenergic sensor binding host epinephrine and norepinephrine. Downstream changes in gene expression are mediated by phosphorylation of its cognate response regulator (RR) QseB, and 'cross-talks' with non-cognate regulators KdpE and QseF to activate motility and virulence. In UPEC, cross-talk between QseBC and TCS PmrAB is crucial in the regulation and phosphorylation of QseB RR that acts as a repressor of multiple pathways, including motility. Here, we investigated QseBC regulation of motility in the atypical Enteropathogenic *E. coli* (EPEC) strain O125ac:H6, causative agent of persistent diarrhoea in children, and its possible cross-talk with the KdpDE and PmrAB TCS. We showed that in EPEC QseB acts as a repressor of genes involved in motility, virulence and stress response, and in absence of QseC HK, QseB is likely activated by the non-cognate PmrB HK, similarly to UPEC. We show that in absence of QseC, phosphorylated QseB activates its own expression, and is responsible for the low motility phenotypes seen in a QseC deletion mutant. Furthermore, we showed that KdpD HK regulates motility in an independent manner to QseBC and through a third unidentified party different to its own response regulator KdpE. We showed that PmrAB has a role in iron adaptation independent to QseBC. Finally, we showed that QseB is the responsible for activation of colistin and polymyxin B resistance genes while PmrA RR acts by preventing QseB activation of these resistance genes.

**Data Availability Statement:** All relevant data are within the manuscript and its Supporting Information files.

**Funding:** This project received funding from the European Union's Horizon 2020 research and innovation programme under the Marie Sklodowska-Curie grant agreement number 765147 (awarded to JMW). The funders had no role in study design, data collection and analysis, decision to publish, or preparation of the manuscript.

**Competing interests:** The authors have declared that no competing interests exist.

## Author summary

Enteropathogenic *Escherichia coli* (EPEC) is a human pathogen and the leading cause of diarrhoea in children under 5 years in low-income countries. EPEC, and specially the sub-group denominated atypical EPEC, has been associated with community-acquired persistent diarrhoea and also is an important agent of post-weaning diarrhoea in young piglets. Two-component systems (TCS) are signalling system used by bacteria to sense and adapt to different stimuli. Bacteria use TCS to detect host signals as cues to exert virulence in their preferred niche. QseBC is a TCS that has been linked to regulation of motility and virulence in other pathogenic *E. coli*. In this study we broaden our understanding of QseBC TCS and its role in regulating virulence traits such as motility in atypical EPEC. We also investigate its capabilities to interact with other TCS, named KdpDE and PmrAB, and how these interactions are responsible for regulation of motility and resistance to antimicrobials such as colistin.

## Introduction

Enteropathogenic *Escherichia coli* (EPEC) is a common human pathogen that adheres to the small intestine and is considered the leading cause of diarrhoea in children under 5 years of age in low-income countries [1]. A recent European study reported a high prevalence of EPEC in samples collected from community-acquired gastroenteritis, suggesting this pathogen is re-emerging [2]. EPEC can also cause diarrhoeal disease in livestock [3] and companion animals [4,5], and can be zoonotic [6] posing an additional burden. EPEC strains are classified into typical (tEPEC) and atypical (aEPEC), depending on the presence of the pEAF virulence plasmid that encodes proteins involved in the formation of the bundle-forming pilus (Bfp) [7]. The Bfp mediates tEPEC adherence to intestinal epithelium and is responsible for the appearance of localised microcolonies on the cell surface otherwise known as a localized adherence (LA) pattern [8]. Due to the absence of *bfp*, aEPEC usually shows a diffused adherence (DA), aggregative adherence (AA) or localized adherence-like (LAL) pattern [9]. Even though tEPEC has been more extensively studied, aEPEC is a more common cause of diarrhoea [10], and is associated with a more persistent diarrheal disease than tEPEC [11,12]. On Spanish farms most cases of post-weaning diarrhoea are associated with enterotoxigenic *E. coli* (ETEC) but this is closely followed by aEPEC. Conversely, aEPEC was more prevalent in cases of diarrhoea in suckling pigs [3].

EPEC shares many similarities with Enterohaemorrhagic *E. coli* (EHEC) [13], associated with foodborne outbreaks that cause bloody diarrhoea in humans of which about 5% lead to haemolytic uremic syndrome (HUS) [14,15]. The main differences between EHEC and EPEC are the ability of EHEC to produce Shiga-Toxin and colonize the large intestine. EHEC and EPEC share a pathogenicity island encoding the locus of enterocyte effacement (LEE) which encodes a type 3 secretion system (T3SS) that injects different effectors into the mammalian intestinal host cell [16], causing microvilli effacement, intimate attachment to the host cell [17], actin pedestal formation, loss of epithelial tight junctions and apoptosis, all which leads to severe diarrhoea [18].

EPEC needs to sense different host signals to exert virulence in their preferred niche. One of the main signalling systems associated to virulence in both EPEC and EHEC is the QseBC two-component system (TCS) [19]. QseBC is a typical TCS comprising the histidine kinase (HK) QseC and the response regulator (RR) QseB [20]. In EHEC, QseC HK has been reported to detect adrenergic signals present in the intestinal tract (epinephrine and norepinephrine),

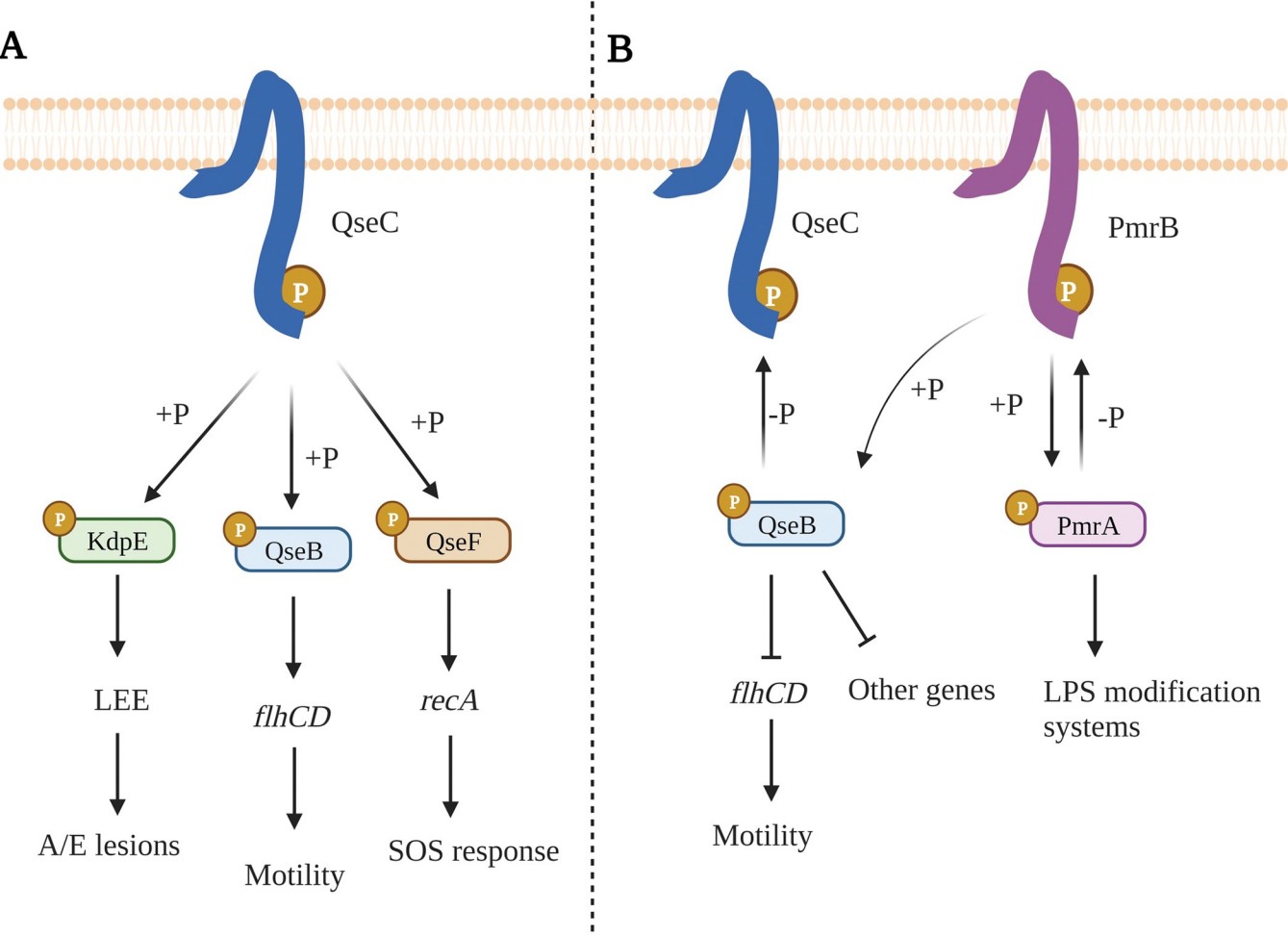

**Fig 1. Simplified theories of QseBC gene regulation in EHEC (A) and UPEC (B). A.** In EHEC, QseC phosphorylates three different RRs KdpE, QseB and QseF that in turn upregulate LEE, *flhC* and *recA* genes involved in A/E lesion formation, motility and SOS response, respectively [23]. **B.** In UPEC, it is theorized that QseB RR acts as a repressor of motility genes and other genes. QseB RR is phosphorylated by the non-cognate HK PmrB and de-phosphorylated by QseC that controls QseB repressor activity [25]. Created with Biorender.com.

and a novel bacterial quorum sensing molecule AI-3 and upon detection induce virulence and motility [21]. Regulation of virulence and motility by QseBC has been extensively studied using knock-out mutants. Deletion of QseC HK decreased motility, virulence [20,22] and expression of *recA* which is important for regulation of the SOS response [23]. This led to the hypothesis that QseC acts as an activator of virulence and motility, as well as other pathways. This classical theory of QseBC regulation (Fig 1A) theorized that, upon signal detection, QseC undergoes an autophosphorylation reaction, and in turn phosphotransfer to the cognate RR QseB, which upregulates *flhCD* the master regulator of flagella, activating motility [23]. Failing to replicate Δ*qseC* phenotypes in Δ*qseB* and Δ*qseB*Δ*qseC* deletion mutants and other results led to the hypothesis that unphosphorylated QseB acts as a repressor of *flhCD* operon [23,24]. Other studies proposed that QseC also phosphorylates the non-cognate RRs KdpE to activate *ler* expression, and QseF to upregulate SOS response and Shiga-toxin expression [23] (Fig 1A).

In Uropathogenic *E. coli* (UPEC) a different mechanism for QseBC regulation of motility and virulence was proposed by Guckes *et al* [25] involving cross-regulation with the PmrAB TCS. PmrAB detects extracellular iron [26] and is involved in resistance to polymyxin B and

colistin [27] in *E. coli* and *Salmonella enterica*. Guckes and co-workers hypothesized that the phenotypes observed in the Δ*qseC* mutant are due to upregulation and over-activation of QseB, that acts as a repressor of motility and other pathways. The authors showed that the PmrB HK can phosphorylate but not dephosphorylate QseB RR *in vitro* [25]. Conversely, QseC was shown to de-phosphorylate QseB *in vitro* and to also sequester it, so it is not available for regulation [28]. This model predicts that deletion of the QseC HK will lead to phosphorylation of QseB RR by the non-cognate PmrB HK, leading to auto-induced expression of QseB and repression of the motility regulator *flhCD*. A Δ*qseC*Δ*pmrB* mutant was shown to recover the WT phenotype, fitting with the idea that in a Δ*qseC* background PmrB phosphorylates QseB RR [25] (Fig 1B).

Due to the avirulent and non-motile phenotypes of a QseC mutant, QseC has been proposed as a good anti-virulence target [29]. Sperandio´s group developed LED209 as inhibitor of the QseC sensing domain which had a similar phenotype to a Δ*qseC* mutant [22]. However, for the further development of inhibitors of TCS as anti-virulence targets, it is important to have a good understanding of the cross-regulatory networks in different *E. coli* phenotypes [30]. Therefore, in this study, we explored the role of QseBC in virulence in atypical Enteropathogenic *E. coli* strain O125ac:H6 [31]. We also investigated potential cross-talk between QseBC and the TCSs PmrAB and KdpDE to regulate motility and expression of virulence genes. We found that, in aEPEC O125:H6, QseBC and PmrAB TCSs cross-regulate motility and virulence genes, and this crosstalk is the responsible of the aberrant phenotypes seen in an aEPEC QseC mutant as previously suggested by Guckes *et al* [25]. We found that KdpDE has a role in motility that does not involve cross-talk with QseBC TCS. Furthermore, we showed that in aEPEC colistin resistance is mediated via crosstalk between QseBC and PmrAB. Based on these results we suggest a theory for QseBC regulation in Enteropathogenic *E. coli* in which QseB acts as a master repressor of motility and virulence and activator of colistin resistance and is tightly regulated via interactions with the TCS PmrAB.

## Results

### Genetic manipulation of atypical EPEC str. O125ac:H6 using a two-plasmid CRISPR-Cas system

Atypical Enteropathogenic *E. coli* strain O125ac:H6 (DSM8711) was acquired from the Leibniz Institute DSMZ collection located in Germany. The presence of the restriction and modification system (HsdR) typically found in clinical strains of *E. coli* [32] was checked by PCR using primers pBM001+pBM002 and shown to be absent. The absence of this restriction-modification system in strain DSM8711 makes it a good candidate for genetic modification of atypical *E. coli*. To generate 6 individual mutants of the two-component system genes Δ*qseB*, Δ*qseC*, Δ*pmrA*, Δ*pmrB*, Δ*kdpD* and Δ*kdpE* we used the two-plasmid CRISPR-Cas system developed by Jiang *et al* [33]. Of particular interest was the cross-regulation between QseBC, reported to have a main role in virulence, and PmrAB and KdpDE, reported to interact with QseBC in UPEC and EHEC, respectively. To study these interactions, we constructed 10 double gene deletion mutants (Δ*qseB*Δ*qseC*, Δ*pmrA*Δ*pmrB*, Δ*pmrA*Δ*qseB*, Δ*pmrA*Δ*qseC*, Δ*pmrB*Δ*qseB*, Δ*pmrB*Δ*qseC*, Δ*kdpD*Δ*qseB*, Δ*kdpD*Δ*qseC*, Δ*kdpE*Δ*qseB* and Δ*kdpE*Δ*qseC*).

For each construct a guide was cloned into pTargetF and a 1kb repair template created by PCR as described in Material and Methods. All templates were designed to introduce in-frame deletions to avoid polar effects on expression of the remaining genes in the operon. For the response regulator gene mutants (*qseB*, *pmrA* and *kdpE*) at least 80% of both the receiver and DNA-binding domain were deleted. For the histidine kinase genes (*qseC*, *pmrB* and *kdpD*) around 1 kb was deleted from the cytoplasmic domain that contains the catalytic domain and

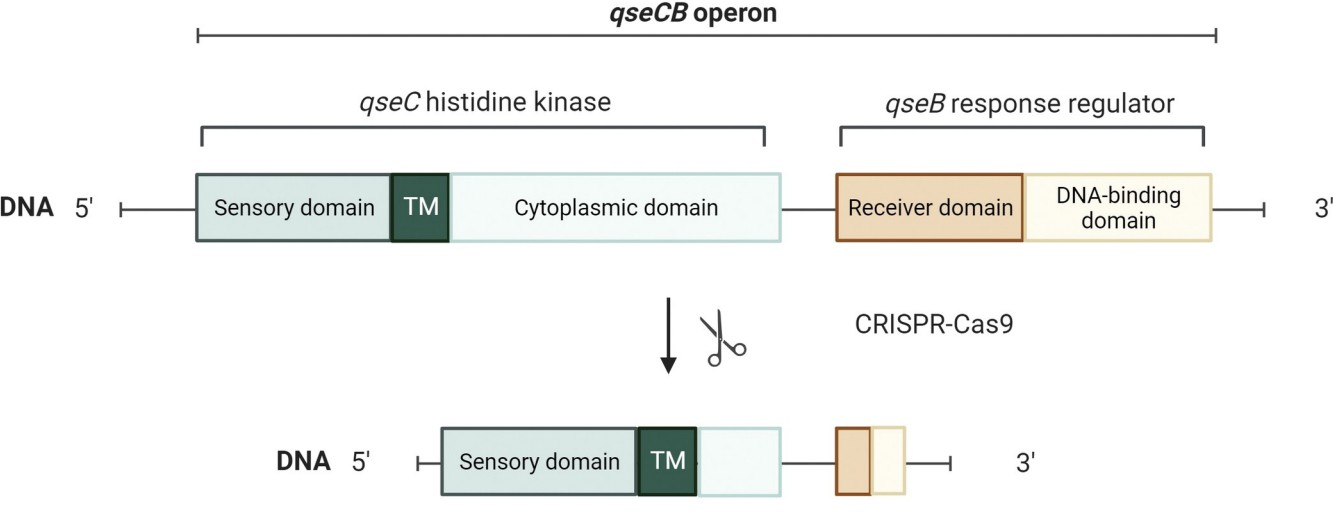

**Fig 2. The strategy used for constructing gene deletions showing the histidine kinase (QseC) and response regulator (QseB) deletion as an example.** In histidine kinases an in-frame deletion was inserted to eliminate more than 80% of the cytoplasmic domain that contains the catalytic domains. Sensory domain and transmembrane region were left functional. More than 80% of the response regulators were deleted, including parts of both the receiver domain and DNA-binding domain. In-frame deletions were used to avoid polar effects on expression of the rest of the operon. Created with Biorender.com.

other important residues for autophosphorylation (Fig 2), ensuring that neither phosphorylation nor de-phosphorylation of the response regulator by HKs would be possible. The sensory and transmembrane domains of HK were left intact.

## Deletion of histidine kinase *qseC* slows growth rate in late-exponential phase

Most of the mutations do not affect growth in LB media compared to the WT control (Fig 3A and 3B). However, four mutants, $\Delta qseC$ (p<0.0001), $\Delta pmrA\Delta qseC$ (p<0.0001), $\Delta kdpD\Delta qseC$ (p<0.0001) and $\Delta kdpE\Delta qseC$ (p<0.0001) showed a delay in growth in late-exponential phase. Growth constants and standard deviation can be found in supporting S1 Table. As all affected mutants contain $\Delta qseC$ mutation, absence of QseC HK is most likely responsible for this phenotype. However, growth of $\Delta pmrB\Delta qseC$ was not affected.

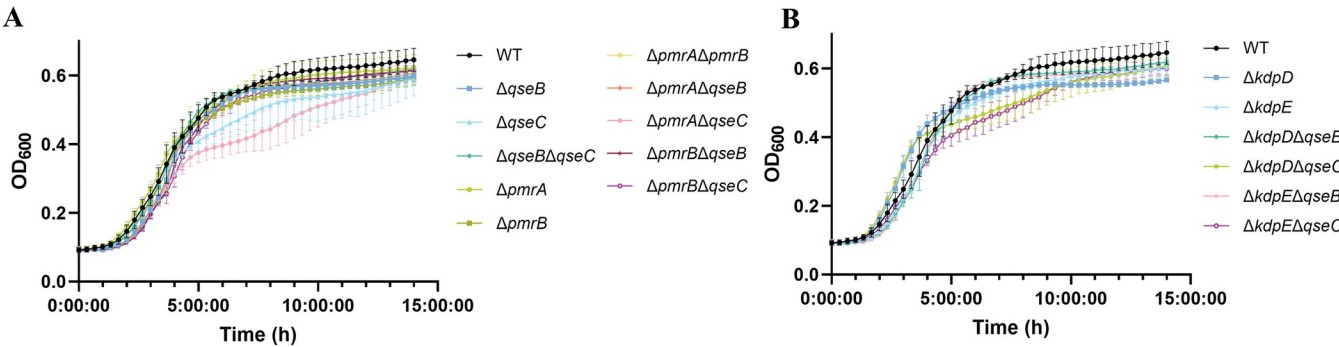

**Fig 3. Growth curves of EPEC O125ac:H6 deletion mutants compared to the WT.** Growth curves represent $OD_{600}$ measurements recorded every 30 minutes for 15 hours for EPEC O125ac:H6 WT, $\Delta qseB$, $\Delta qseC$, $\Delta qseB\Delta qseC$, $\Delta pmrA$, $\Delta pmrB$, $\Delta pmrA\Delta pmrB$, $\Delta pmrA\Delta qseB$, $\Delta pmrA\Delta qseC$, $\Delta pmrB\Delta qseB$, $\Delta pmrB\Delta qseC$ **(A)** and $\Delta kdpD$, $\Delta kdpE$ $\Delta kdpD\Delta qseB$, $\Delta kdpD\Delta qseC$, $\Delta kdpE\Delta qseB$ and $\Delta kdpE\Delta qseC$ **(B)**. Error bars represent the Standard Deviation, n = 4.

## Response regulator QseB is a key repressor of motility and is activated by the histidine kinase PmrB in a Δ*qseC* mutant

The soft agar motility assay was used to determine the individual contributions of QseBC and PmrAB as well as their potential interactions on bacterial motility. Deletion of the QseB RR, alone and in combination with QseC HK, PmrA RR and PmrB HK leads to a significant increase in motility compared to the WT, indicating that QseB acts as a repressor of motility. Single and double mutants from PmrAB TCS show similar motility to the WT, which suggests that this TCS alone is not involved in regulating motility (Fig 4).

The Δ*qseC* mutant shows a reduction of motility. The deletion of both QseC HK and PmrA RR leads to even lower motility than the single QseC mutant. However, *E. coli* Δ*pmrB*Δ*qseC*, with simultaneous deletion of the QseC and PmrB HKs, recovers the WT phenotype (Fig 4A). These data are consistent with the hypothesis that QseB RR acts as a repressor of motility and is activated by phosphorylation via PmrB in the QseC mutant. To prove that off-target effects are not responsible for the observed phenotypic changes we complemented the Δ*qseB* strain with *qseB* WT gene, leading to WT motility phenotypes (Fig 4B). As hypothesized Δ*pmrB*Δ*qseC* mutant complemented with *pmrB* WT gene has reduced motility due to absence of *qseC*. In a second step, complementation with WT *qseC* gene restored the WT phenotypes (Fig 4B). These gene complementation studies show that off-target effects are not responsible for the changes in phenotypes seen in the mutants.

Finally, to study the contribution of phosphorylation in the cross-regulation between QseBC and PmrAB, we constructed non-phosphorylatable mutants of QseB and PmrB. QseBD51A shows a similar phenotype to WT indicating that the presence of unphosphorylated QseB is sufficient to repress motility to WT levels. PmrBH152A would be unable to phosphorylate QseB. Introduction of PmrBH152A in a Δ*qseC* mutant restores WT motility phenotypes, indicating that PmrB is likely responsible for phosphorylation of QseB in a Δ*qseC* mutant (Fig 4B). Introduction of the non-phosphorylatable QseBD51A into a Δ*qseC* background also restores WT levels of motility, corroborating the hypothesis that phosphorylation of QseB is responsible of the aberrant phenotypes in the Δ*qseC* mutant (Fig 4B).

## The two-component system KdpDE does not crosstalk with QseBC in the regulation of motility

KdpDE TCS has also been reported to interact with QseBC to regulate virulence in EHEC. Motility data of single and double mutants is shown in Fig 5. Absence of KdpD HK lowers motility significantly compared to the control, however, it is not comparable to the lack of motility showed by the QseC mutant. Double mutants Δ*kdpD*Δ*qseB* and Δ*kdpE*Δ*qseB*, and Δ*kdpD*Δ*qseC* and Δ*kdpE*Δ*qseC* show similar phenotypes to Δ*qseB* and Δ*qseC*, respectively, suggesting that there is no cross-regulation between these TCSs in the regulation of motility. Finally, KdpE RR single mutant shows similar motility to the WT, indicating that KdpD regulation of motility is through a third unidentified party.

## Motility of atypical *E. coli* is affected by iron but not by adrenergic signals or AI-3 in absence of the PmrAB TCS

QseBC signalling is activated by adrenergic hormones, such as epinephrine, norepinephrine and the autoinducer-3 [21,23] (Fig 6A), while PmrAB is activated by extracellular iron [26]. We explored the effects of these molecules and iron on EPEC O125ac:H6 WT motility by including different concentrations (500 nM, 5 μM, 50 μM and 500 μM) of each in the soft agar. In contrast to other findings reported in the literature [34,35], no significant change in motility

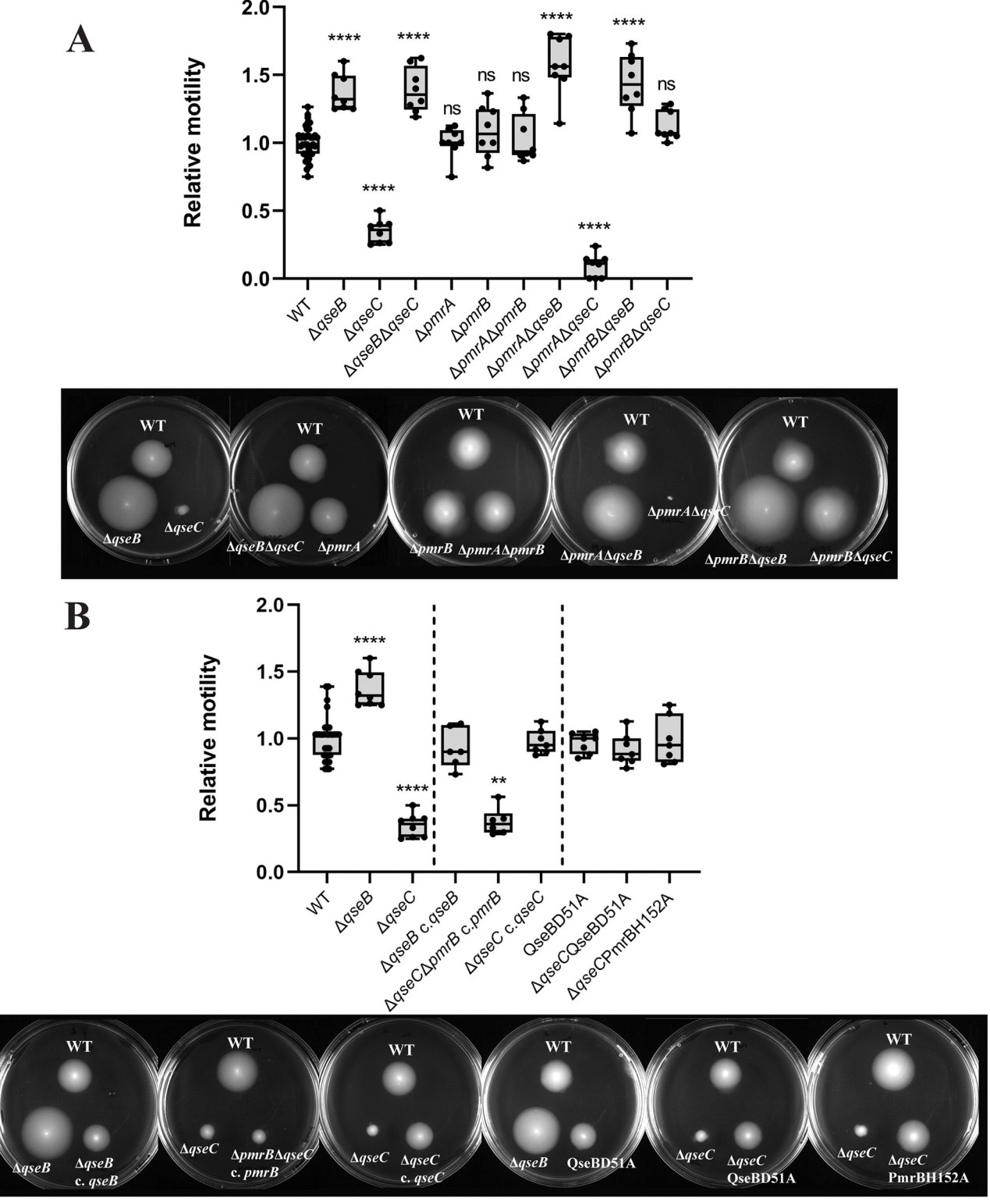

**Fig 4. Relative motility of QseBC and PmrAB single and double mutants. A.** Relative motility compared to the WT of EPEC O125ac:H6 Δ*qseB*, Δ*qseC*, Δ*qseB*Δ*qseC*, Δ*pmrA*, Δ*pmrB*, Δ*pmrA*Δ*pmrB*, Δ*pmrA*Δ*qseB*, Δ*pmrA*Δ*qseC*, Δ*pmrB*Δ*qseB*, Δ*pmrB*Δ*qseC*. **B.** Relative motility compared to the WT of EPEC O125ac:H6 Δ*qseB* complemented with WT *qseB*, Δ*pmrB*Δ*qseC* complemented with WT *pmrb* Δ*qseC* complemented with *qseC*, WT QseBD51A, Δ*qseC* QseBD51A and Δ*qseC* PmrBH152A. WT motility is calculated as the relative motility of the WT per plate compared to the average motility of the WT in the experiment. Motility plates that exemplify the seen motility are depicted. * p-value <0.05; ** <0.01; *** <0.001; **** <0.0001 via one-way ANOVA with Tukey's multiple comparison (A) and via Krukal-Wallis test with Dunn's multiple comparison (B), n = 8.

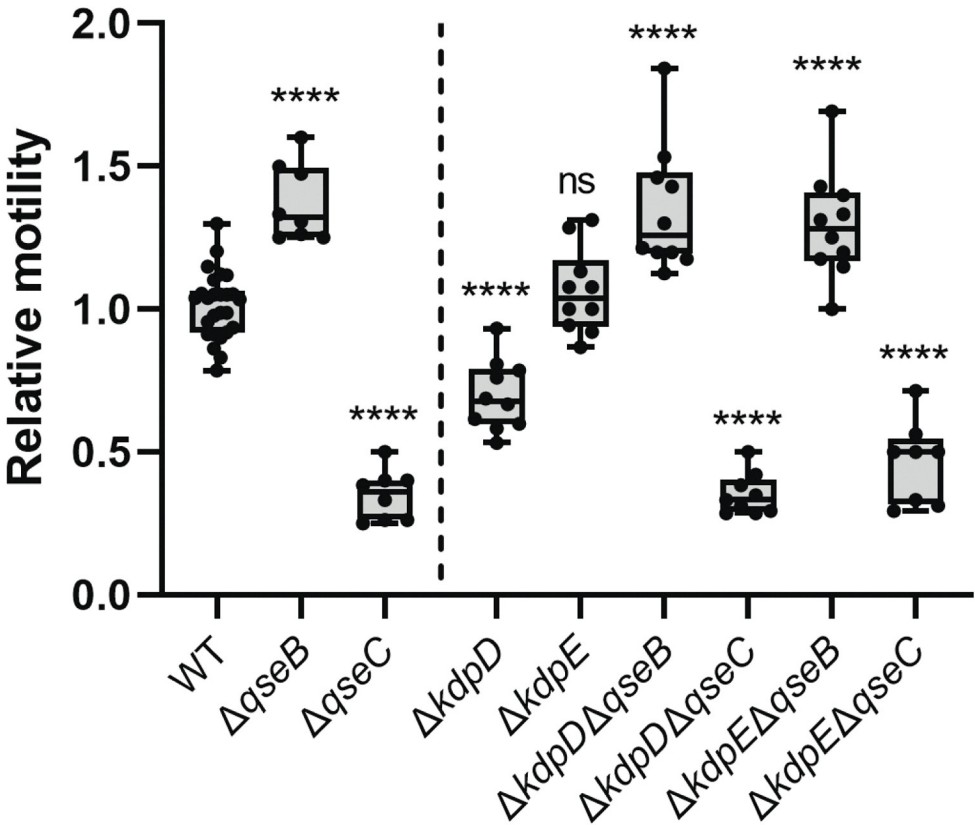

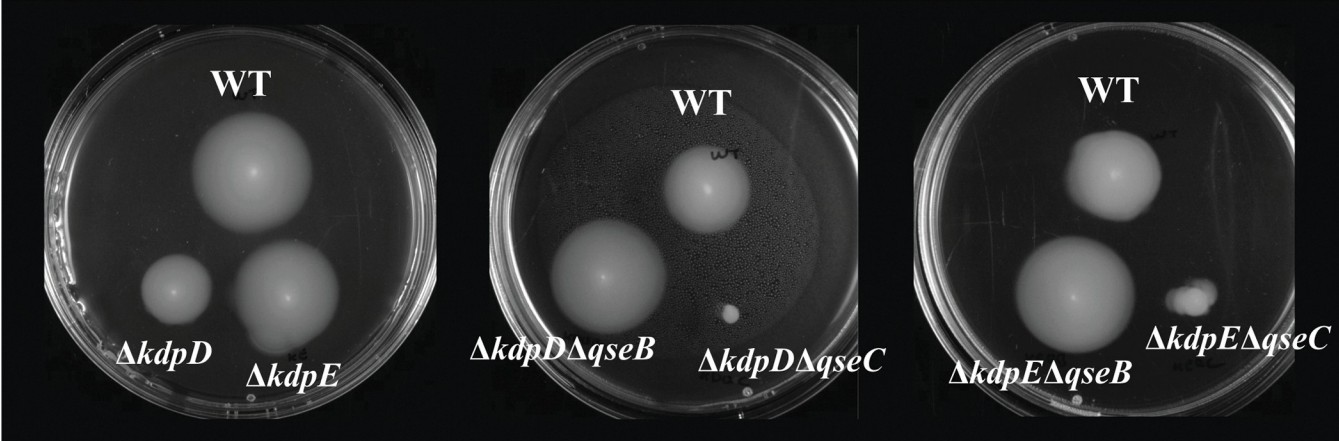

**Fig 5. Relative motility of QseBC and KdpDE single and double mutants.** Relative motility compared to the WT of eight replicates is depicted for EPEC O125ac:H6 *ΔqseB*, *ΔqseC*, *ΔkdpD*, *ΔkdpE ΔkdpDΔqseB*, *ΔkdpDΔqseC*, *ΔkdpEΔqseB* and *ΔkdpEΔqseC*. EPEC O125ac:H6 WT motility is calculated as the relative motility of the WT per plate compared to the average motility of the WT in the experiment. Motility plates that exemplify the seen motility are depicted. * p-value <0.05; ** <0.01; *** <0.001; **** <0.0001 via one-way ANOVA with Tukey's multiple comparison, n = 8.

was observed with epinephrine at any of the tested concentrations. AI-3 and $Fe^{3+}$ also did not affect motility of the WT (Fig 6B).

We then explored the effects of epinephrine, AI-3 and $Fe^{3+}$ on the motility of single mutants from QseBC and PmrAB (*ΔqseB*, *ΔqseC*, *ΔpmrA* and *ΔpmrB*). No differences were found for epinephrine and AI-3. However, presence of $Fe^{3+}$ significantly reduced the motility of *ΔpmrA*

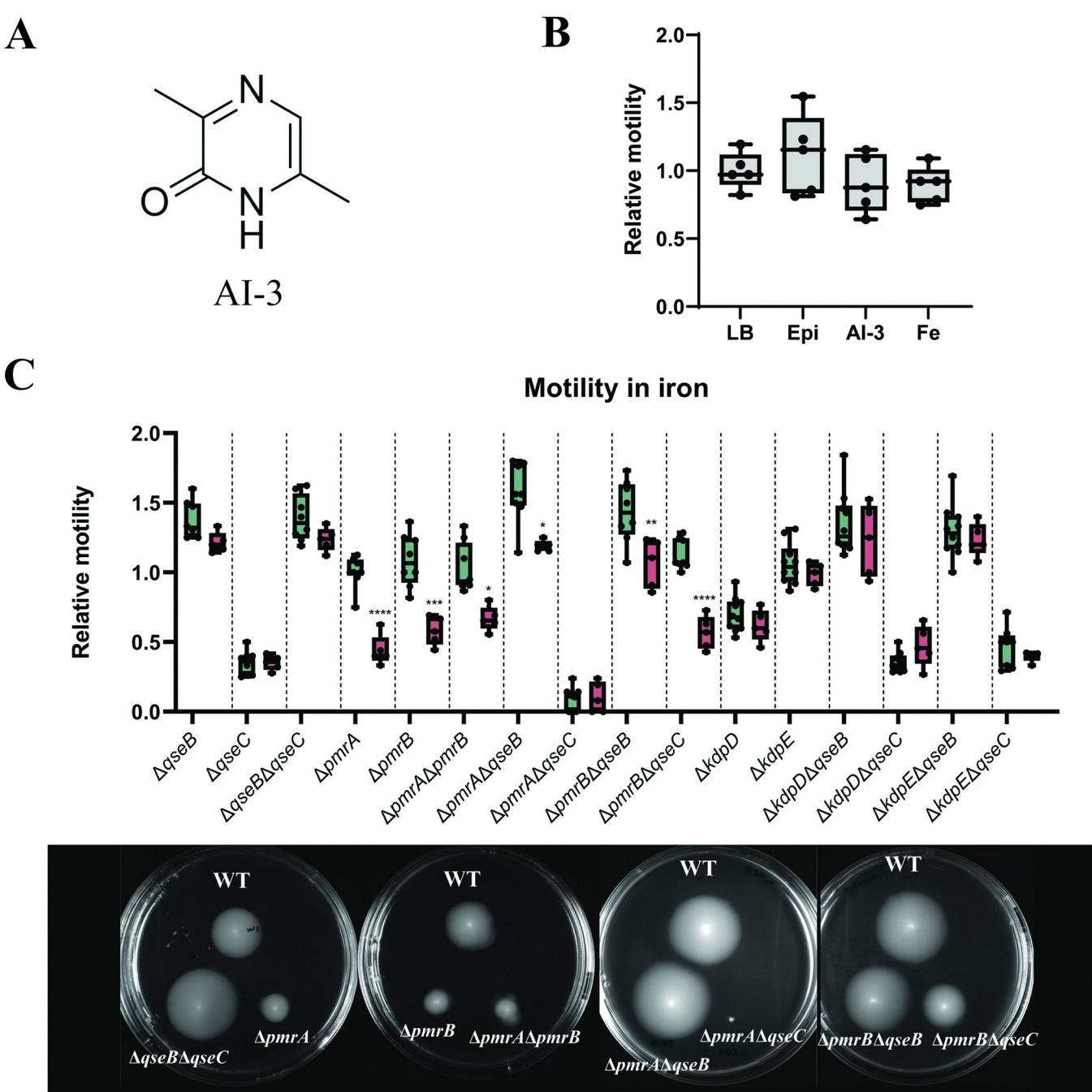

**Fig 6. Effect of adrenergic signals (epinephrine), AI-3 and iron in the motility of EPEC O125ac:H6. A**. Chemical structure of AI-3 [36] **B.** Relative motility compared to the non-treated control of EPEC O125ac:H6 in presence of 500 μM of epinephrine, AI-3 and $Fe^{3+}$. **C**. Comparison between the relative motility of EPEC O125ac:H6 mutants compared to the WT in non-treated agar plates (green) with the relative motility of EPEC O125ac:H6 mutants compared to the WT in agar plates containing 500 μM $Fe^{3+}$ (red). Motility plates of mutants that are affected by the presence of iron are depicted. * p-value <0.05; ** <0.01; *** <0.001; **** <0.0001 via one-way ANOVA with Tukey's multiple comparisons, n = 5.

and Δ*pmrB* compared to the non-treated control. Motility of all mutants was then assayed in presence of 500 μM of $Fe^{3+}$ (Fig 6C). Reduction of relative motility in the presence of iron was seen with mutants Δ*pmrA*, Δ*pmrB*, Δ*pmrA*Δ*pmrB*, Δ*pmrA*Δ*qseB*, Δ*pmrB*Δ*qseB* and Δ*pmrB*Δ*qseC*. This data suggests that PmrAB TCS has an independent role in adapting to

metal ion stress that does not involve QseBC TCS, and in presence of excess iron, PmrAB is needed for maximum motility.

## Differential gene expression using luciferase reporter assays, confirms cross-regulation of flagella and other virulence genes by QseBC and PmrAB TCS

To investigate the effects of TCS mutants on expression of genes associated with motility (*flhC* and *fliA*), LEE island expression (*ler*, main transcriptional regulator of the LEE island [37]) and regulation of the SOS response (*recA*) we constructed a luciferase reporter of the cognate promoters as explained in Material and Methods. *ler* expression was evaluated in DMEM medium containing low amounts of glucose, due to its poor expression in LB medium. The *bla* gene was used as a promoter control, as it was not expected to be regulated by TCSs.

Expression of all genes (except *bla* control), *fliA*, *flhC*, *ler* and *recA* was upregulated in Δ*qseB*, Δ*qseB*Δ*qseC*, Δ*pmrA*Δ*qseB* and Δ*pmrB*Δ*qseB* in LB media (Fig 7A). Upregulation of the flagella regulatory genes matches with the observed increased motility (Fig 4). Expression of all tested genes in Δ*pmrA*, Δ*pmrB*, Δ*pmrA*Δ*pmrB* Δ*kdpD* and Δ*kdpE* was similar to WT expression. Mutant Δ*pmrB*Δ*qseC* also had a similar expression to WT in all tested genes, indicating that introduction of PmrB HK mutation into Δ*qseC* (HK) recovers both WT phenotype and gene expression. Finally, Δ*qseC* and Δ*pmrA*Δ*qseC* show significant downregulation of flagella regulatory genes *flhC* and *fliA* and the stress regulator *recA*. In LB medium, the expression of *ler* in Δ*qseB* mutants is increased, indicating that in this condition QseB acts as a repressor. However, *ler* is not differently regulated compared to the WT in low glucose DMEM in any of the tested mutants, indicating that in activated conditions QseBC does not play a major role in regulation. To confirm this data we performed a qPCR in Δ*qseB* and Δ*qseC* mutants of *ler* and other related virulence gene *espA* (Fig 7B) in low glucose DMEM. Non-significant difference in expression with respect to the WT was found. This is different to what was reported in literature for EHEC [23]. Detailed graphs showing differential expression can be found in supporting data S1 Fig and raw luminescence data in S2 Table.

To check the validity of the promoter reporter assay we checked expression of *flhC* and *fliA* genes in the Δ*qseB* and Δ*qseC* mutants using qPCR. We also included Δ*pmrA*Δ*qseC* because there was a significant decrease in luminescence with the *bla* promoter in this mutant compared to the WT. In the Δ*qseB* mutant *flhC* and *fliA* were upregulated but only *fliA* was significant. As expected, both genes were downregulated in the Δ*qseC* and Δ*pmrA*Δ*qseC* mutants (Fig 7C), confirming the luminescence data. Differences in upregulation and downregulation seen between the luminescence reporter and qPCR data are probably due to the collection of culture for qPCR during mid-exponential phase, representing only one time point, while the luminescence reporter assay allows for kinetic measurements.

Finally, we investigated *qseB* expression in Δ*qseC* and Δ*pmrA*Δ*qseC* using qPCR. QseB RR is highly upregulated in both mutants (around 100-times upregulation) (Fig 7D), supporting the hypothesis that *qseB* upregulation is one of the reasons for the aberrant phenotypes seen in Δ*qseC* and Δ*pmrA*Δ*qseC*.

## *E. coli* Δ*pmrA*Δ*qseC* is resistant to colistin and polymyxin B due to constitutive upregulation of colistin resistance regulon

PmrAB has been reported to regulate colistin and polymyxin B resistance in *E. coli* and other Gram-negative bacteria [38]. To test whether cross-regulation between PmrAB and QseBC TCSs would also affect expression of colistin and polymyxin B resistance genes we performed MIC assays with the single and double mutants. The double mutant Δ*pmrA*Δ*qseC* has a MIC

**A**

| Mutant | *flhC* | *fliA* | *ler* | *recA* | *bla* |
|---|---|---|---|---|---|
| Δ*qseB* | + | ++ | + | + | . |
| Δ*qseC* | − | −− | . | − | . |
| Δ*qseB*Δ*qseC* | + | ++ | + | + | . |
| Δ*pmrA* | . | . | . | . | . |
| Δ*pmrB* | . | . | . | . | . |
| Δ*pmrA*Δ*pmrB* | . | . | . | . | . |
| Δ*pmrA*Δ*qseB* | + | ++ | + | + | . |
| Δ*pmrA*Δ*qseC* | − | −− | . | − | − |
| Δ*pmrB*Δ*qseB* | + | ++ | + | + | . |
| Δ*pmrB*Δ*qseC* | . | . | . | . | . |
| Δ*kdpD* | . | . | . | . | . |
| Δ*kdpE* | . | . | . | . | . |

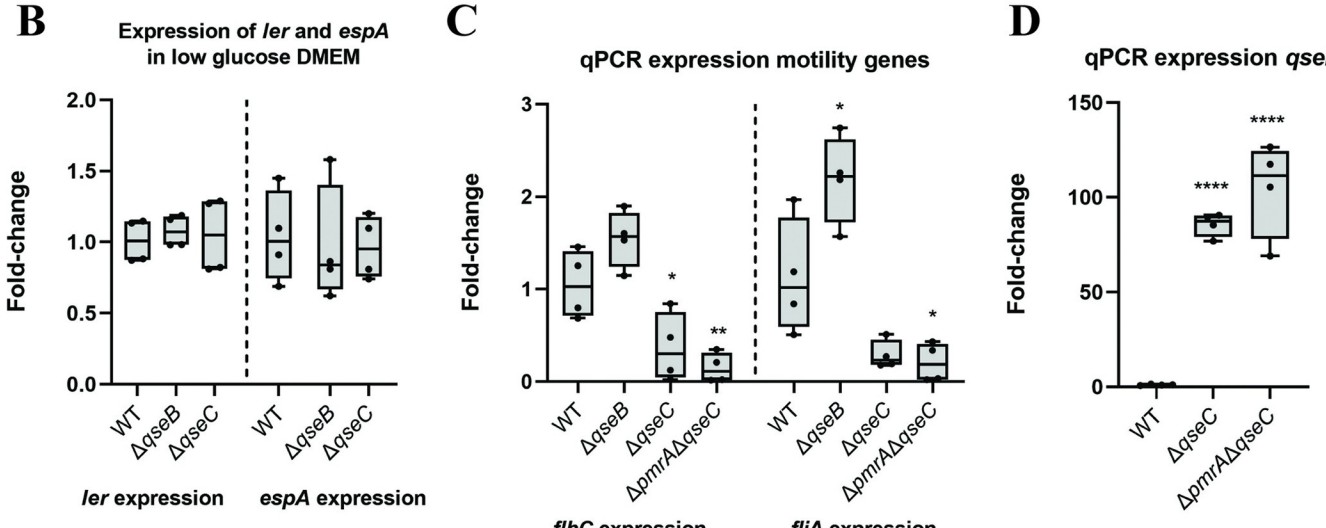

**Fig 7. Differential expression of *flhC*, *fliA*, *ler*, *recA* and *qseB* in different EPEC O125ac:H6 QseBC, PmrAB and KdpDE mutants. A.** Differential expression according to the luciferase assay of the genes *flhC*, *fliA*, *ler*, *recA* and *bla* of all tested mutants. Significant differences are indicated by ++ > 2-fold; + 1-2-fold;. non-significant differences with WT; - 1–0.5-fold and—- < 0.5-fold **B.** Quantitative PCR of genes *ler* and *espA* in low glucose DMEM in EPEC O125ac:H6 H6 Δ*qseB* and Δ*qseC* **C.** qPCR of genes *flhC* and *fliA* in EPEC O125ac:H6 Δ*qseB*, Δ*qseC*, and Δ*pmrA*Δ*qseC* to prove that luciferase assay was accurate. **D.** qPCR in EPEC O125ac:H6 Δ*qseC* and Δ*pmrA*Δ*qseC* to check *qseB* expression in these mutants. * p-value <0.05; ** <0.01; *** <0.001; **** <0.0001 via one-way ANOVA with Tukey's multiple comparisons, n = 4.

(4 μg/ml)16-times higher than the WT (0.25 μg/ml). The MIC of all the other mutants was 0.25 μg/ml for both colistin and polymyxin B, same as the WT (Fig 8A). As the PmrAB regulon includes, amongst others, the proteins EptA and ArnB, involved in LPS modification pathways

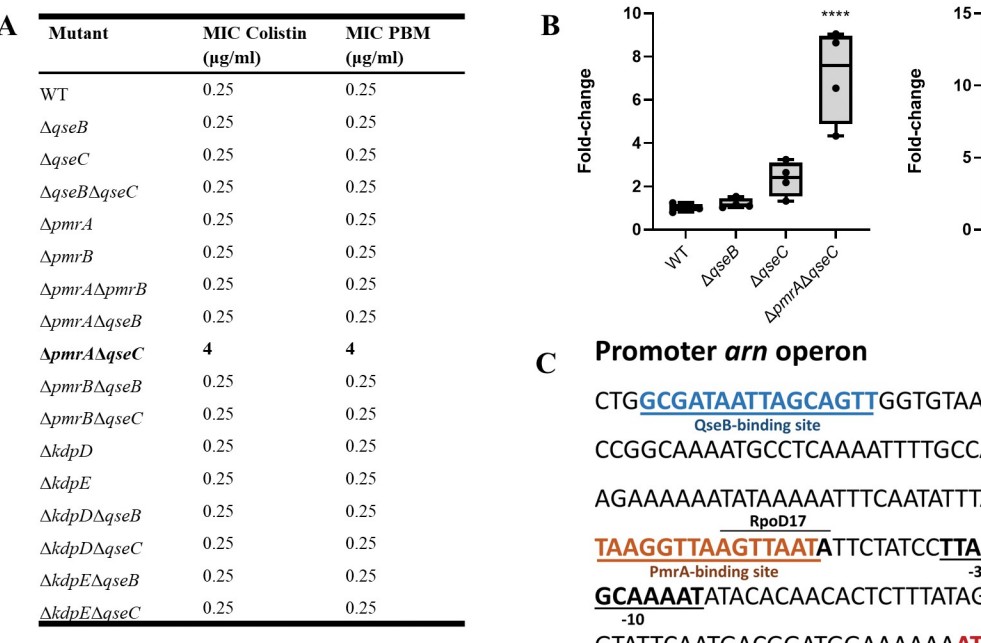

**A**

| Mutant | MIC Colistin (µg/ml) | MIC PBM (µg/ml) |
|---|---|---|
| WT | 0.25 | 0.25 |
| ΔqseB | 0.25 | 0.25 |
| ΔqseC | 0.25 | 0.25 |
| ΔqseBΔqseC | 0.25 | 0.25 |
| ΔpmrA | 0.25 | 0.25 |
| ΔpmrB | 0.25 | 0.25 |
| ΔpmrAΔpmrB | 0.25 | 0.25 |
| ΔpmrAΔqseB | 0.25 | 0.25 |
| **ΔpmrAΔqseC** | **4** | **4** |
| ΔpmrBΔqseB | 0.25 | 0.25 |
| ΔpmrBΔqseC | 0.25 | 0.25 |
| ΔkdpD | 0.25 | 0.25 |
| ΔkdpE | 0.25 | 0.25 |
| ΔkdpDΔqseB | 0.25 | 0.25 |
| ΔkdpDΔqseC | 0.25 | 0.25 |
| ΔkdpEΔqseB | 0.25 | 0.25 |
| ΔkdpEΔqseC | 0.25 | 0.25 |

**C** **Promoter *arn* operon**

CTG**GCGATAATTAGCAGTT**GGTGTAATATTAAAAATCCTATGATG
<u>QseB-binding site</u>

CCGGCAAAATGCCTCAAAATTTTGCCAAATGCAAAGTCTAAATA

AGAAAAAATATAAAAATTTCAATATTTACGTCTAATATTAGTTT**CT**
RpoD17

**TAAGGTTAAGTTAAT**ATTCTATCC**TTAAAA**TTTTGCTCCAAA**TG**
PmrA-binding site                                      -35

**GCAAAAT**ATACACAACACTCTTTATAGCAAATATAAGTGGACAG
-10

GTATTCAATGACGGATGGAAAAAA**ATG**

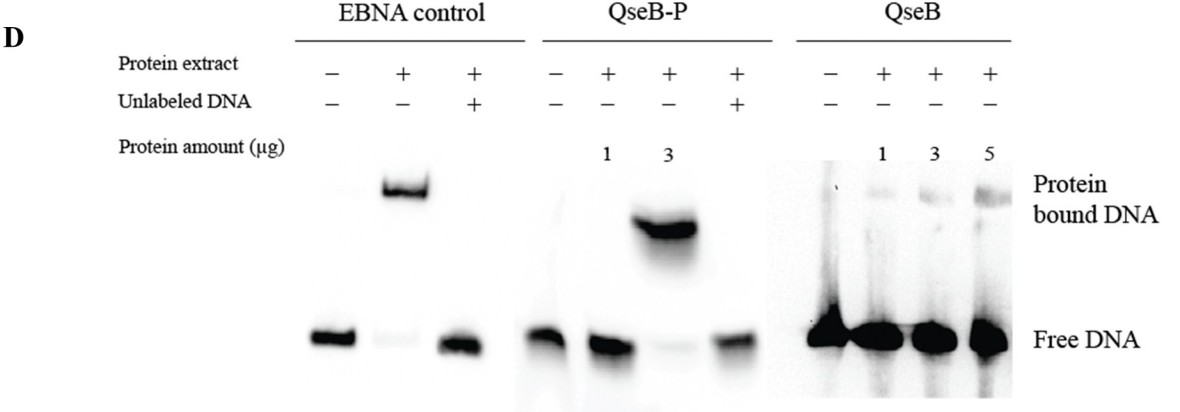

**Fig 8. Effect of mutations in QseBC, PmrAB and KdpDE genes in colistin and polymyxin B resistance. A**. MIC data of colistin and polymyxin B (PBM) for EPEC O125ac:H6 WT, ΔqseB, ΔqseC, ΔqseBΔqseC, ΔpmrA, ΔpmrB, ΔpmrAΔpmrB, ΔpmrAΔqseB, ΔpmrAΔqseC, ΔpmrBΔqseB, ΔpmrBΔqseC., ΔkdpD, ΔkdpE ΔkdpDΔqseB, ΔkdpDΔqseC, ΔkdpEΔqseB and ΔkdpEΔqseC. **B.** Differential expression using qPCR of colistin and PBM resistance genes *arnB* and *eptA* in the mutants EPEC O125ac:H6 WT, ΔqseB, ΔqseC and ΔpmrAΔqseC* p-value <0.05; ** <0.01; *** <0.001; **** <0.0001 via one-way ANOVA with Tukey multiple comparison test, n = 4. **C**. In the promoter region of the *arn* operon, binding motifs are found for both QseB (blue) and PmrA (orange). The -35 and -10 sites were predicted using BPROM [42], and a rpoD17 binding site was predicted within the PmrA binding site. The start (ATG, red) of *arnB* gene is also shown. **D.** Electrophoretic mobility shift assay (EMSA). To validate the technique a EBNA control provided by the kit is shown. Biotylinated DNA fragment containing the QseB binding site in the *arn* operon is found to shif with the presence of 3 µg of QseB-P, which is reversed by the presence of unlabeled QseB site. Finally unphosphorylated QseB is found to bind with lower affinity to the QseB binding site.

[39], we hypothesized that increased resistance was due to upregulation of the PmrAB regulon when PmrA RR and QseC HK are deleted. Quantitative PCR in mutants ΔqseB, ΔqseC and ΔpmrAΔqseC showed that indeed genes *arnB* and *eptA* are constitutively upregulated in ΔpmrAΔqseC compared to the WT control, explaining increased colistin and polymyxin B

resistance (Fig 8B). Upon inspection of the regulatory region of *arn* operon in the genome of *E. coli* O125ac:H6, we identified the consensus binding sequence of PmrA [(C/T)TTAA(G/T)-N5-(C/T)TTAA(G/T)] [40] with a slight modification, as well as the consensus binding sequence of QseB [G(C/T)(G/A)ACAAT(A/T)A-N4-TT] [24,41] also with a slight modification (Fig 8C). This suggests that both QseB and PmrA are implicated in regulating expression of the *arn* operon. To prove this hypothesis we performed an electrophoretic mobility shift assay with phosphorylated and non-phosphorylated QseB and a DNA fragment containing the predicted QseB binding site in the *arn* operon. We found that 3 μg of phosphorylated QseB completely shifts the DNA fragment containing the predicted binding site, indicating that phosphorylated QseB binds indeed to this region (Fig 8D). Non-phosphorylated QseB seems to bind the QseB site with lower affinity (Fig 8D).

## Discussion

In this study, we investigated the crosstalk between the three two-component systems QseBC, PmrAB and KdpDE in atypical Enteropathogenic *E. coli* strain O125ac:H6, and how this crosstalk affects motility and expression of relevant genes. KdpDE seems to have a role in motility independent of QseBC. Motility data of KdpD HK and KdpE RR mutants suggests that KdpD HK activates motility via an alternative response regulator to the cognate KdpE. Similar expression of *flhC* and *fliA* in the KdpD HK mutant to the WT suggests that KdpDE regulation of motility is likely through control of flagella late-regulatory genes. Unlike what was seen in EHEC [23], KdpDE does not seem to be involved in regulation of LEE in aEPEC.

We showed that inactivation of QseC HK results in lack of motility in aEPEC, as reported for EHEC, UPEC and other Gram-negative bacteria such as *Salmonella enterica* [20,29,35,43], which was consistent with previous studies suggesting that QseC was an activator of motility [44]. However, introduction of a PmrB HK deletion into Δ*qseC* background, resulted in similar motility to the WT strain, confirming the hypothesis suggested by Guckes *et al* [25] that QseB RR acts as repressor of motility and, in the absence of QseC HK, QseB is phosphorylated by the non-cognate PmrB HK.

In aEPEC, unlike what was reported in UPEC and EHEC, deletion of the response regulator *qseB* alone and in combination with Δ*qseC*, Δ*pmrA* and Δ*pmrB* leads to increased motility and expression of genes *flhC*, *fliA*, *ler* and *recA*. This confirms that the QseB response regulator acts as repressor of multiple genes. Introduction of a non-phosphorylatable QseB variant (QseBD51A) in aEPEC maintained WT motility phenotypes indicating that basal repression exerted by QseB RR is likely due to unphosphorylated QseB binding with low affinity to QseB targets, leading to weak repression of these genes.

Both Δ*qseC* and Δ*pmrA*Δ*qseC* show lack of motility and downregulation of *flhC*, *fliA* and *recA*, together with high upregulation of *qseB* expression, likely responsible for these phenotypes. Over-expression of *qseB* is due to phosphorylated QseB autoregulating its own expression. Our hypothesis is that, in the WT, QseB can be phosphorylated by the non-cognate PmrB HK, but it is immediately de-phosphorylated or sequestered by QseC [25,28] (Fig 9A). In the absence of QseC HK, phosphorylated QseB induces its own expression and an excess of activated QseB in the cell in turn represses multiple genes involved in motility, stress response and probably more unidentified genes (Fig 9B). The introduction of QseBD51A in a Δ*qseC* background leads to WT phenotypes, suggesting that phosphorylation of QseB in the absence of QseC is responsible for the aberrant phenotypes. The introduction of PmrBH152A in the Δ*qseC* background also leads to WT phenotypes, suggesting that this phosphorylation occurs through PmrB. Further *in vitro* protein phosphorylation and de-phosphorylation studies with the aEPEC protein variants of PmrAB and QseCB would help to confirm this hypothesis. It is

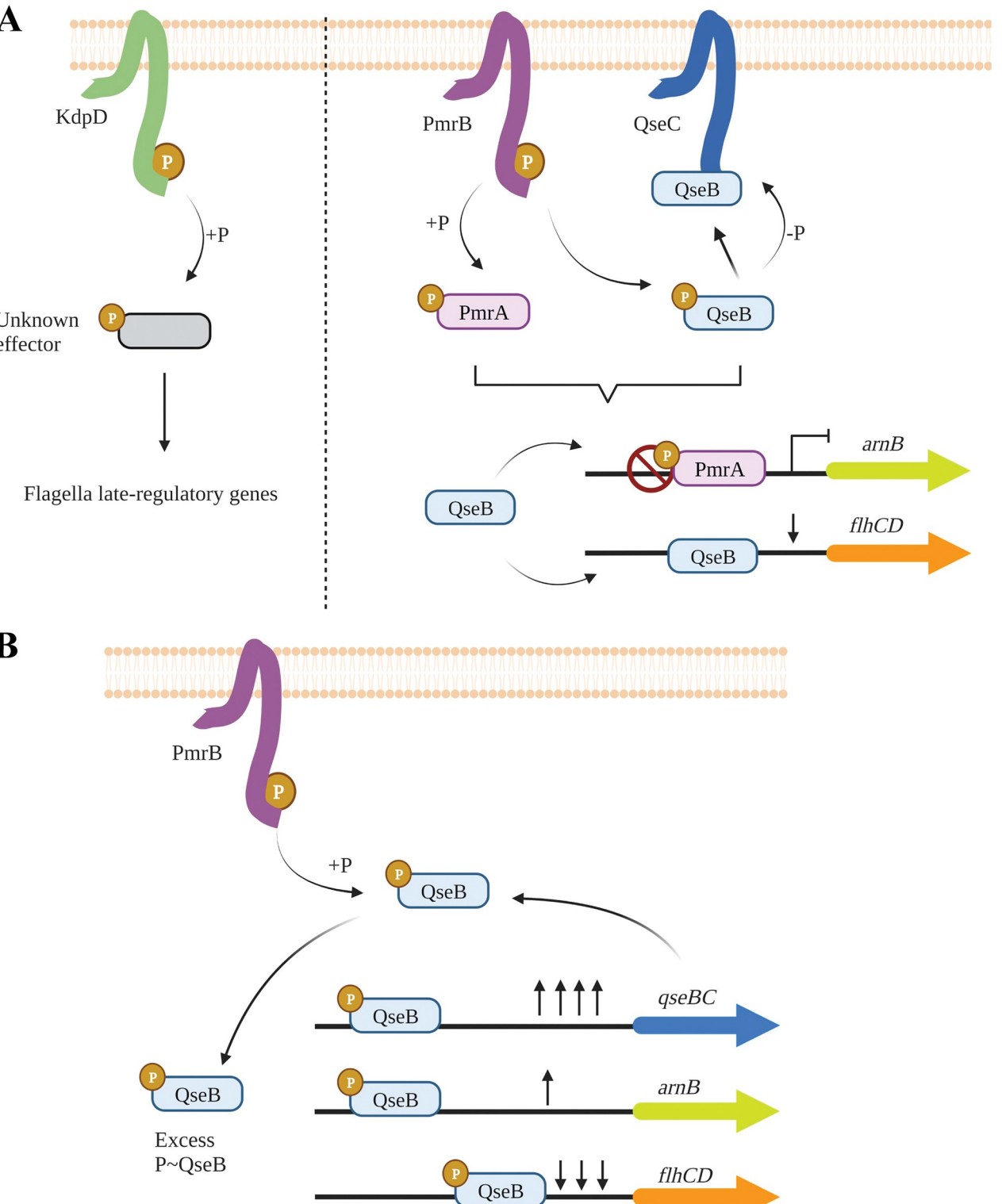

**Fig 9. Proposed regulation of the TCS KdpDE, PmrAB and QseBC in EPEC. A**. In WT aEPEC, KdpD HK phosphorylates an unknown effector to activate flagella late-regulatory genes. PmrB activates its cognate RR PmrA, and the non-cognate QseB. QseC HK controls the level of activated QseB via de-phosphorylation and sequestration of activated QseB. Unphosphorylated QseB in EPEC WT is able to exert a basal repression of *flhC*, *recA* and *ler* genes. **B**. In Δ*pmrA*Δ*qseC* mutant, QseB is phosphorylated via PmrB, and cannot be deactivated by QseC. P~QseB activates the expression of *qseBC* operon leading to QseB over-expression. In *arn* operon, PmrA usually blocks transcription, but in its absence phosphorylated QseB leads to activation of colistin resistance genes. Excess of phosphorylated QseB tightly represses *flhC* and *recA* genes. Created with Biorender.com.

unknown if the positive feedback loop caused by QseB phosphorylation aims to increase repression via QseB, or to increase *qseC* expression that in turn would deactivate the cognate QseB. In the Δ*pmrB*Δ*qseC* mutant, QseB cannot be phosphorylated leading to a recovery of WT phenotypes.

Interestingly, *ler* regulator is upregulated in the Δ*qseB* mutant in LB media but *ler* expression is similar to the WT in the Δ*qseC* mutant. However, when the experiments were performed in low glucose DMEM all the mutants showed a similar expression to the WT. We confirmed this data by qPCR of two genes involved in adhesion to mammalian cells, *ler* and *espA*. We suggest that QseB is only able to repress *ler* expression in certain conditions. The tight regulation of LEE island by multiple factors may explain why in DMEM low glucose QseBC plays a minor role in regulation. Using the FIMO tool [45] from MEME Suite [46] with the binding motifs of PmrA and QseB we found that in the *E. coli* O125ac:H6 genome 16 genes harbour both binding sites in the promoter region including *flhCD*, *arnB*, *ler* and *qseB*. This indicates that these regulators directly regulate *ler* and *flhCD* expression as showed before in literature [24]. Other regulated genes such as *fliA* or *recA* do not contain the motifs suggesting indirect regulation. In fact, *fliA* is probably upregulated through the master regulator of flagella *flhCD* [47].

In a further confirmation of these TCSs interaction, we saw that the Δ*pmrA*Δ*qseC* mutant is 16-times more resistant to polymyxin B and colistin than the WT due to constitutive upregulation of colistin resistance regulon. This increase in resistance was not seen in any of the other mutants. The presence of binding sites for both QseB and PmrA in *arn* operon suggests that both play a role in regulation. Direct regulation was confirmed for QseB using an EMSA assay. We suggest that PmrA binds to the *arn* promoter, blocking binding of the RNA polymerase and the initiation of the transcription (Fig 9A). Only when PmrA is not bound, and QseB is upregulated and activated via PmrB, the *arn* operon is expressed. (Fig 9B). Taken together, all these findings suggest that PmrA and QseB act together to tightly regulate certain genes and QseB can act as activator or repressor depending on the target gene.

We did not find aEPEC to respond with higher motility to adrenergic signals, such as epinephrine, nor the bacterial signalling molecule AI-3. However, we did find that, in presence of excess $Fe^{3+}$, strains containing mutations in either Δ*pmrA* or Δ*pmrB* show significantly lower motility than the WT. This data indicates that in absence of PmrAB, bacteria cannot respond correctly to excess iron. In fact, PmrAB has been linked to adaptation to excess environmental iron [27] and mutations in PmrAB have also been shown to increase sensitivity to low pH in excess iron [48]. In Uropathogenic *Escherichia coli* it has been shown that the iron homeostasis protein Fur leads to motility repression [49]. Thus it is possible that absence of PmrAB prevents iron homeostasis, leading to an excess intracellular iron and an effect of Fur on motility. When the mutations Δ*pmrA* or Δ*pmrB* were combined with either Δ*qseB* or Δ*qseC*, relative motility was lower than the non-treated control, but we still observed higher motility in Δ*pmrA*Δ*qseB* and Δ*pmrB*Δ*qseB* than in the single mutants Δ*pmrA* and Δ*pmrB*. This suggests that the adaptation of motility to excess iron via PmrAB is not related to the QseBC regulation of motility and that PmrAB has an independent role in iron homeostasis.

PmrAB and QseBC are phylogenetically very close to each other, with a 45.83% identity between RRs, and 32.61% identity between the cytoplasmic domains of HKs. In the evolution of these two TCS the sensing domains have diverged (no homology found), such as they sense different stimuli, but the protein sequences involved in the cross-talk between the two HK cytoplasmic domains and RRs have been maintained. PmrAB and QseBC are also present in other enteric Gram-negative bacteria with high degree of identity to the proteins in aEPEC (80–98%), such as in UPEC, EHEC, *Salmonella enterica*, *Shigella* and *Klebsiella pneumoniae*, indicating their importance. Percentages of identity between QseBC and PmrAB from these

bacteria is similar to the one found in aEPEC, and it is possible that cross-regulation occurs in other bacteria than *E. coli*. Point mutations in PmrAB TCS in *Klebsiella* are associated to resistance to colistin [50,51]. It is should be investigated if these point mutations lower affinity of PmrA to promoters of genes involved in colistin resistance, thereby allowing QseB to upregulate colistin resistance. For other Gram-negative bacteria, such as *Acinetobacter* and *Pseudomonas* only the presence of PmrAB TCS has been reported. Here, we demonstrated that regulation in aEPEC and UPEC is similar, however discrepancies have been reported with EHEC regulation found in literature [21,23]. PmrA and QseB regulators are identical in aEPEC and UPEC, and they belong to the B2 phylo-group, EHEC possesses some different amino acids (S2 Fig) and belongs to E phylo-group possibly accounting for the differences in regulation.

In conclusion we suggest that PmrAB and QseBC TCSs may have co-evolved to tightly regulate gene expression. It is unknown in which conditions QseB repression is activated. It is possible that the bacteria uses PmrA and QseB interactions as internal mechanism to regulate expression in a time dependent manner. It is also possible that QseBC signalling has evolved to activate QseB repression only in certain situations where decreased flagella production and stress response are beneficial for the bacteria. Further research is needed to fully understand QseBC and PmrAB regulation. This research also highlights the importance of selecting specific TCS as targets for development of anti-virulence compounds against Gram-negative bacteria.

## Material and methods

### Bacterial strains and growth conditions

Atypical Enteropathogenic *Escherichia coli* O125ac:H6 (DSM8711) was purchased from the Leibniz Institute DSMZ and cultured in Luria-Bertani (LB) medium (Merck Millipore) and when necessary supplemented with kanamycin (50 μg/ml) or spectinomycin (100 μg/ml), and incubated at 37˚C. Chemically competent *Escherichia coli* TOP10 (Invitrogen) was used for cloning. The whole genome sequence of *E. coli* O125ac:H6 was obtained using Illumina-sequencing (MicrobesNG https://www.microbesng.com). Accession number PRJNA930679.

### Construction of deletion mutants, site-directed mutagenesis and mutant complementation using CRISPR-Cas

For the construction of CRISPR-Cas deletion mutants in EPEC O125ac:H6 we used the two-plasmid system developed by Jiang *et al* [33]. First, guides targeting the gene of interests were designed using Benchling software. To introduce the guides in pTargetF, we amplified pTargetF using primer pBM004 that contains a XhoI restriction site, and guide primers (pBM005, pBM013, pBM021, pBM029, pBM037, pBM045) that anneal with the plasmid just after the 20-bp CRISPR guide complementary to the genome. These primers include a SpeI recognition site and the new guide as an overhang. All plasmids and primers used can be found in S3 and S4 Tables. Both pTargetF and amplification product were cut using SpeI FD and XhoI FD (Molecular biology, ThermoScientific), cleaned and ligated using T4 DNA ligase (Promega). Ligation was transformed into chemically competent TOP10 *E. coli* and plated on LB agar containing spectinomycin. Plasmid from colonies growing in spectinomycin was extracted using QIAprep spin miniprep kit (QIAGEN) and introduction of the new guide was confirmed via Sanger sequencing, using primer pBM003 and GATC LightRun tubes services (Eurofins Genomics, Ebersberg, Germany).

The PCR product containing the repair template was constructed using PCR SOEing [52] to generate an amplicon with 500-bp regions of homology upstream and downstream of the region to be deleted. The repair template was designed to retain several amino-acids of the deleted gene resulting in an in-frame deletion. The homologous fragments were amplified by PCR and then purified using Invisorb Fragment CleanUp (Invitek, Berlin, Germany). The homologous repair template was generated by SOEing PCR using 50 ng of each homologous region as template and external primers for amplification and was purified using Invisorb Fragment CleanUp kit.

To complement the mutations Δ*qseB* and Δ*pmrB*Δ*qseC*, we used CRISPR-Cas9 to introduce the WT gene (amplified using primers pBM007+pBM010 for *qseB*, pBM007+pBM010 for *qseC* and pBM007+pBM010 for *pmrB*). Guides used for complementation can be found in S4 Table. Finally, to construct the non-phosphorylatable mutants we used CRISPR-Cas9 to change the conserved aspartic acid 51 to alanine in QseB and the conserved histidine 152 to alanine in PmrB. To do this a CRISPR guide containing the amino-acid of interest was selected. Construction of the templates for the non-phosphorylatable variant was performed using primers pBM0072+pBM0073 and pBM075+pBM0076 that contain an overhang correspondent to the selected guide where silent mutations were introduced in the codons to make it different enough from the guide for the Cas9 not to cut after homologous recombination.

To construct the mutants, electrocompetent cells of EPEC O125ac:H6 were prepared as reported by Jiang *et al* [33], and transformed by electroporation with 100 ng of the temperature sensitive plasmid pCas (0.2 cm-gap cuvettes (Bio-Rad), in a Gene pulser Xcell electroporation system (Bio-Rad)). After electroporation 1 ml SOC media (Invitrogen) was added to the cuvettes and transferred to 1.5 ml Eppendorfs. Cells were left to recover at 30°C for 3 hours at shaking 180 r.p.m. and then plated in LB kanamycin (50 μg/ml) plates and incubated at 30°C. Colonies containing pCas were selected by colony PCR using the primers pBM053 and pBM054.

Electrocompetent cells of EPEC O125ac:H6 pCas were prepared as previously reported with slight modifications [33]. Cells were grown in 50 ml tubes with LB and 50 μg/ml kanamycin at 30°C with shaking at 200 r.p.m. to $OD_{600}$ 0.1, at which point arabinose was added to a final concentration of 10 mM to induce the λ-red recombination system. After induction, bacteria were grown to $OD_{600}$ 0.5, pelleted by centrifugation at 4°C, washed twice with cold sterile demi-water and stored in 200 μl aliquots cold 12% glycerol. 50 μl aliquots of electrocompetent cells were incubated for 30 minutes with 100 ng pTargetF with the guide of interest and 500 ng of template and electroporated. After 3-hour incubation at 30°C cells were plated in LB plates with kanamycin and spectinomycin and incubated at 30°C. Deletion mutants were selected by colony PCR using primer annealing outside the designed template.

To curate the mutants of pTargetF, the bacteria were grown in LB medium containing kanamycin and 0.5 mM IPTG at 30°C to induce expression of a *pMB1* guide present in pCas that targets pTargetF plasmid. Loss of pTargetF was confirmed by streaking the culture in kanamycin and spectinomycin plates. After elimination of pTargetF, pCas was cured by growing the mutants at 37°C, the non-permissive temperature for replication. Loss of pCas was confirmed by plating of the cultures in kanamycin plates. The PCR to confirm gene deletion was then repeated.

## Growth curve

Growth curves of the different mutants were performed by adding 200 μl of a 1:100 dilution into 96-well plates. Condensation was avoided by adding 1 ml anti-condense agent (0.05% Triton X-100 in 20% ethanol) to the lid of the plate and leaving it to dry. Incubation at 37°C was

performed in SpectraMax M5 (Molecular Devices LLC, San Jose, CA, USA) and $OD_{600}$ measured every 30 minutes for 15 hours. Experiments were repeated four independent times.

## Motility assays

Motility assays were performed using a soft agar plate motility assay. Briefly, motility plates were made by pouring 20 ml of LB soft agar (LB+0.3% agar (Alfa Aesar, Haverhill, MA, USA)) into 90x16 mm Petri dishes (VWR, Radnor, PA, USA). When necessary, agar was pre-mixed with 500 nM, 5 μM, 50 μM or 500 μM epinephrine (Sigma-Aldrich), AI-3 or $FeCl_3$. Experiments with epinephrine were protected from the light at all times to prevent oxidation. After solidification, plates were left to dry for 15 minutes. Single colonies from the mutant of interest were picked with a 10 μl pipette tip and then poked into the motility plate. Plates were incubated for 18 hours at 37˚C, and motility measured in cm with a ruler. We assayed the motility of two mutants and the WT per motility plate. Motility of each mutant was calculated as the relative motility compared to the WT on the same plate. WT variability was accounted by measuring the relative motility of the WT in each plate compared to the average motility of the WT on different plates from the same experiment. Pictures of the motility plates were made using GelDoc XR+ gel documentation system (Bio-Rad). Experiments were performed 8 times.

## AI-3 synthesis

Fig 10 shows the synthetic scheme of autoinducer AI-3. Starting material (3-chloro-2,5-dimethylpyrazine) was purchased from commercial vendor (Enamine, Frankfurt, Germany) and used without additional purification.

## Construction of luciferase reporter plasmids

For the construction of the luciferase reporter plasmid, first we amplified the *luxAB* luciferase genes from *Photorhabdus luminescens* from pXen-1 plasmid [53] using the primers pBM055+-pBM056 that contain a SpeI and EcoRI site overhang respectively. This was cloned between the SpeI and EcoRI sites of pTargetF so that *luxAB* was under the control of the constitutive promoter J23119 that in pTargetF is flanked by BamHI and SpeI sites, creating in this way pTarget_lux plasmid.

The promoters located upstream the genes of interest *flhC*, *fliA*, *bla*, *ler* and *recA* were amplified by PCR (≈1 kb fragments) using the genomic DNA of EPEC O125ac:H6 as template, except for *bla* that was amplified form plasmid pXen-1. The ampicillin resistance gene *bla* gene was selected as a promoter control. To assure correct positioning of the promoter fragments the primers were designed to include a few amino-acids from the gene of interest that would be in-frame with the luciferase gene after annealing. The primers (pBM059 to

**Fig 10. Synthetic scheme of autoinucer AI-3.**

pBM068) contain BamHI and SpeI overhangs to facilitate replacement of the constitutive promoter in pTarget_lux. After cloning the promoter fragments into pTarget_lux, the insertion of the promoter was checked by colony PCR using the primers pBM057 and pBM058.

## Differential expression using luciferase assay

Mutants were transformed by electroporation with the luciferase reporter plasmids: pTargeF_*flhC*, pTargetF_*fliA*, pTargetF_*bla*. pTargetF_*ler*, pTargetF_*recA*. For the differential expression assay, 200 μl of 1:100 dilution in LB+spectinomycin or low glucose DMEM+-spectinomycin for pTargetF_*ler* (GIBCO) of the mutants with the reporter plasmid were transferred to a clear bottom 96-well plate (Corning Incorporated). To avoid cross-luminescence the wells around each culture were left empty. As substrate for the luciferase, 50 μl of 1% nonanal (Sigma-Aldrich) in mineral oil (Sigma-Aldrich) [54] was added to all the gaps between the wells. Evaporation was avoided by spreading 1 ml of anti-condense agent to the lid and letting it dry. Plates were incubated at 37°C in a SpectraMax M5 reader, and luminescence was measured every 30 minutes for 18 hours. Experiments were repeated in quadruplicate.

The peak-luminescence (highest luminescence found during the 18 hours) was recorded. Per promoter, relative luminescence in each experiment was calculated relative to the average of the WT. WT variability was calculated as the luminescence of individual WT measurements relative to the average luminescence for the WT in the experiment.

## Differential expression using qPCR

EPEC O125ac:H6 WT, EPEC O125ac:H6 Δ*qseB*, EPEC O125ac:H6 Δ*qseC* and EPEC O125ac:H6 Δ*qseC*Δ*pmrA* were grown in 6 ml of LB media to $OD_{600}$ 0.4 or 10 ml low glucose DMEM to $OD_{600}$ 0.2. Cultures were centrifuged at 4°C, 4.000 r.p.m. for 10 minutes, supernatant discarded, and pellets frozen using liquid nitrogen. RNA extraction from the pellets was performed using RNeasy Mini Kit (QIAGEN) following manufacturer´s manual. RNA concentration was measured using Qubit RNA broad range kit (Invitrogen) in a Qubit 4 fluorometer (Invitrogen). QuantiTect reverse transcription kit (QIAGEN) was used for cDNA synthesis from 1 μg RNA. Quantitative PCR was performed using GoTaq qPCR Master Mix (Promega) and primers shown in S5 Table. *gyrA* was chosen as control gene. Differential expression was calculated using the $2^{-\Delta\Delta Ct}$ method [55]. Experiments were repeated 4 times.

## Minimal inhibitory concentration (MIC) assay

Minimal inhibitory concentrations against EPEC O125ac:H6 were performed as described by European Committee on Antimicrobial Susceptibility Testing [56]. Briefly, double dilutions of colistin (Duchefa Biochemie, Haarlem, The Netherlands) and polymyxin B (Sigma-Aldrich) were performed in Mueller Hinton broth (MHB) horizontally in a 96-well plate over a range of concentrations 16–0.015 μg/ml in a final volume 100 μl. A 1:100 dilution of the desired mutant was performed and 100 μl added to the plate for a final volume of 200 μl. Plates were incubated overnight at 37°C. The next day, $OD_{600}$ was measured using SpextraMax M5 and MIC recorded as the minimal concentration preventing any visible bacterial growth.

## Electrophoretic mobility shift assay (EMSA)

The QseB response regulator cloned in pNIC28-Bsa4 [57] plasmid containing $His_6$-tag was expressed and purified in *E. coli* BL21(DE3) using His-affinity (GE Healthcare) and size exclusion chromatography. EMSA assays were performed using the LightShift Chemiluminescent EMSA kit (Thermo Scientific). First, 20 μg of QseB were phosphorylated by addition of 0.1 M

of lithium potassium acetyl phosphate (Sigma-Aldrich) in 50 mM TrisCl, 100 mM KCl and 10 mM MgCl$_2$ and incubated at room temperature for 45 minutes. Biotinylated primers in the 5' region from the QseB-binding site in the *arn* promoter region were purchased from Integrated DNA Technologies (Leuven, Belgium) (Fw-/5BiosG/ATTTACGTCTAATATTAGTTTCTT AAGGTTAAGTTAATATTCTATCCTTAAAATTTTGCT; Rv-/5BiosG/AGCAAAATTTTA AGGATAGAATATTAACTTAACCTTAAGAAACTAATA TTAGACGTAAAT). Oligos were annealed by mixing equimolar concentrations in TE buffer (10 mM Tris-Cl, 1 mM EDTA) and heating to 95˚C followed by cooling at 1˚C per second in a thermocycler. Binding reactions were performed by adding 0. 1 or 3 μg of phosphorylated or non-phosphorylated QseB, and 20 fmol of the biotinylated QseB-site region. Controls provided by the kit and a control with non-biotylinated QseB site (200 pmol) were included in the assays. Binding reactions were loaded with the provided loading dye in 8% polyacrylamide in 0.5X TBE (Thermofisher) gels. A mini trans-blot cell (Bio-Rad) was used to transfer proteins and DNA to Biodyne B positively charged nylon membranes (ThermoFisher) for 45 minutes at 100 V. Cross-link was performed by exposing the membranes to UV light for 15 minutes in a GelDoc XR+ (Bio-Rad). Blocking, incubation with streptavidin-horseradish peroxidase conjugate, washing and revealing were performed according to the protocol. Membranes were imaged using ChemiDoc XRS+ (Bio-Rad).

### Statistical analysis

Growth curves were fitted to a logistic growth and the rate of growth was calculated per each curve (n = 3 per mutant). A one-way ANOVA with Tukey's multiple comparison test was performed to test significance. Motility (n = 8) and gene expression data (n = 4) were first confirm to be normal using a Shapiro-Wilk test. Significance was tested using a one-way ANOVA with Tukey's multiple comparison test. When data was not normal a Kurksal-Wallis test with Dunn's multiple comparison was used (adjusted p-value<0.05 considered significant).Statistical analysis was performed using Prism 9 (Graphpad Software, San Diego, USA).

## Supporting information

**S1 Table. Growth rate of WT and *Escherichia coli* O125ac:H6 mutants together with standard deviation (SD).** Growth rate was calculated by fitting growth curves into a logistic growth equation (n = 3).
(PDF)

**S2 Table. Raw luminescence data expressed in RIL x100.** Individual values of four replicates are represented in the peak-luminescence of each strain and each gene.
(PDF)

**S3 Table. Plasmids used in this study.**
(PDF)

**S4 Table. Primers used in this study for cloning.**
(PDF)

**S5 Table. Primers used in this study for qPCR.**
(PDF)

**S1 Fig. Differential expression of *flhC*, *fliA*, *ler*, *recA* and *bla* in different EPEC O125ac:H6 QseBC, PmrAB and KdpDE mutants.** Differential expression according to the luciferase assay of the genes *flhC*, *fliA*, *ler* in LB or low glucose DMEM and *recA* of all tested mutants is depicted (A, B, C, D, E and F). * p-value <0.05; ** <0.01; *** <0.001; **** <0.0001 via one-

way ANOVA with multiple comparisons, n = 4.
(TIF)

**S2 Fig. QseB and PmrA alignment between atypical Enteropathogenic *Escherichia coli* O125ac:H6, Uropathogenic *Escherichia coli* UTI89 and Enterohemorrhagic *Escherihica coli* O157:H7.** Light blue shows identical amino-acids, dark blue shows changes in the amino-acid sequence.
(TIF)

## Author Contributions

**Conceptualization:** Blanca Fernandez-Ciruelos, Jerry M. Wells.

**Funding acquisition:** Jerry M. Wells.

**Investigation:** Blanca Fernandez-Ciruelos, Tasneemah Potmis.

**Resources:** Vitalii Solomin.

**Supervision:** Jerry M. Wells.

**Writing – original draft:** Blanca Fernandez-Ciruelos.

**Writing – review & editing:** Blanca Fernandez-Ciruelos, Jerry M. Wells.

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
