## [Decision Letter · Decision Letter 0]

6 Jun 2023

Dear Prof. dr. Wells,

Thank you very much for submitting your manuscript "Cross-talk between QseBC and PmrAB two-component systems is crucial for regulation of motility and colistin resistance in Enteropathogenic *Escherichia coli*" for consideration at PLOS Pathogens. As with all papers reviewed by the journal, your manuscript was reviewed by members of the editorial board and by three several independent reviewers. In light of the reviews (below this email), we would like to invite the resubmission of a significantly-revised version that takes into account the reviewers' comments.

The reviewers came to the conclusion that your paper is of interest for publication in Plos Pathogens but they raised some issues to be adressed. After reading the reviews and looking at the manuscript, I recommend Major Revision based on the critiques from reviews. I am sorry I cannot be more positive at the moment, however we are looking forward to receiving your revision. Note that we may send your paper back to some of the more critical reviewers upon resubmission.

You will have to reply point by point to the reviewers comments, but I would like you to pay particular attention to the following suggestions and give them due consideration:

1) Genetic and in vitro phosphorylation are required for supporting the link between PmrB and QseB

­2) DNA-protein experiments are necessary to complete the regulation model that propose the authors

­3) Experiments regarding LEE expression in the in vitro conditions turn to be not erformed as experimentally required

­4) Can you justify why is it important to study the effect of adrenergic signals for the pathophysiology of EPEC infection?

5) Studying colistin resistance is interesting in the context described in this paper. However, you should justify/explain such a goal in a paper deciphering TCS in EPEC. For instance, should we understand that EPEC are likely to become colistin resistance (and more likely than other E. coli)?

We cannot make any decision about publication until we have seen the revised manuscript and your response to the reviewers' comments. Your revised manuscript is also likely to be sent to reviewers for further evaluation.

Sincerely,

Thomas Guillard, PharmD, PhD

Academic Editor

PLOS Pathogens

David Skurnik

Section Editor

PLOS Pathogens

Kasturi Haldar

Editor-in-Chief

PLOS Pathogens

orcid.org/0000-0001-5065-158X

Michael Malim

Editor-in-Chief

PLOS Pathogens

orcid.org/0000-0002-7699-2064

I am returning your manuscript with three reviews. The reviewers came to the conclusion that your paper is of interest for publication in Plos Pathogens but they raised some issues to be adressed. After reading the reviews and looking at the manuscript, I recommend Major Revision based on the critiques from reviews. I am sorry I cannot be more positive at the moment, however we are looking forward to receiving your revision. Note that we may send your paper back to some of the more critical reviewers upon resubmission.

You will have to reply point by point to the reviewers comments, but I would like to pay particular attention to the following suggestions and give them due consideration:

1) Genetic and in vitro phosphorylation are required for supporting the link between PmrB and QseB

­2) DNA-protein experiments are necessary to complete the regulation model that propose the authors

­3) Experiments regarding LEE expression in the in vitro conditions turn to be not erformed as experimentally required

­4)Can you justify why is it important to study the effect of adrenergic signals for the pathophysiology of EPEC infection?

5) Studying colistin resistance is interesting in the context described in this paper. However, you should justify/explain such a goal in a paper deciphering TCS in EPEC. For instance, should we understand that EPEC are likely to become colistin resistance (and more likely than other E. coli)?

Reviewer's Responses to Questions

**Part I - Summary**

Reviewer #1: Fernández-Ciruelos et al describe the transcriptional talk between the QseBC and PmrAB two- component systems in enteropathogenic Escherichia coli (EPEC). This interaction regulates the motility and colistin resistance. The authors show genetic phenotypic evidence, however, the direct regulation of motility and colistin resistance genes is lack. QseB-P-DNA experiments are required for supporting the observations that QseB-P directly control these two regulons.

1. Does the in vitro and in vivo QseB phosphorylation in the absence of QseC is PmrB-dependent?

2. Should be analyzed which motility genes (fliC, flhDC, among others) are affected by the transcriptional talk. Is there a consensus site for QseB-P in the motility genes? Does QseB-P bind to these promoter regions?

3. Should be corroborated if the QseB-binding site found for arn (and eptA) is recognized by QseB-P (site-directed mutagenesis). This analysis also applies for motility genes.

Reviewer #2: In this work, Fernandez-Ciruelos et al, take a comprehensive approach to test which of the two current QseBC regulation models apply to atypical enter-pathogenic E. coli (aEPEC). This goal is significant, because it recognizes the extensive diversity across E. coli pathotypes that is too often simplified in the literature. The authors create a suite of deletion mutants in a representative aEPEC strain and then test how each mutant swims, or responds to colistin or adrenergic molecules. They show that aEPEC follows the profiles reported for UPEC, where QseB serves as a repressor of motility and its likely activation involves cross-regulation by PmrB. The analyses are robust, powered and provide convincing genetic evidence that support the authors' conclusions. Likewise, the authors provide transcriptional data to demonstrate down regulation of motility genes and go on to show that QseBC and PmrAB modulate resistance to colistin. Again, the experiments are powered and justify the conclusions. There has been substantial debate in the field as to which model of QseBC regulation is correct. Here - in the case of aEPEC - it appears that the observations made in UPEC also apply to aEPEC.

Some minor suggestions to strengthen the paper:

1) The authors provide no complementation in their studies. They should consider complementing with wild-type QseB or a phosphomimetic to validate their results.

2) In EHEC and UPEC, QseB varies by a single amino acid and swapping the two does not alter QseB actvity. It would be good to provide the QseB and QseC sequence alignments between UPEC, EHEC and aEPEC.

3) One thought is that phylogeny may dictate QseBC regulation. Is the particular aEPEC strain used in E, A, or B2 clade?

4) Please provide the statistical test used for each analysis and in each figure. Just stating that GraphPad 9 was used is inadequate.

Reviewer #3: This is an interesting study to uncover the role of three TCS: QseBC, KdpDE and PmrAB in regulation of motility, virulence and antimicrobial resistance in atypical EPEC. They show that crosstalk between QseBC and PmrAB at the level of QseB phosphorylation controls flagella expression and motility. They also show that KdpD independently from KdpE contraols motility. They suggest that QseB, but not QseC, and KdpE control LEE expression, and show that both PmrAB and QseBC systems control antimicrobial resistance.

Strenghs is the cool and important cross talk between these three TCSs.

1) The construction of an impressive collection of single and double mutants.

2) The flagella henotypes and motility studies are very well conducted.

3) The role of iron regulation through pmrB is novel

4) The egulation of antimicorbial resistance is also very novel and important.

5) These studies have been conducetd in a atypical EPEC that has not been passage through the lab for very long

In terms of weaknesses some controls are missing, and in experiments regarding LEE expression the in vitro conditions are wrong.

1) Either coplementation of the mutants and a quick motility assay, or alternatively whole genome sequencing of them is needed. It is important to ensure that there were no accumulation of secondary mutations during genetic manipulation (very common in E. coli) and that the phenotypes can indeed be atributed to the genes mutated.

2) Growth curves on Fig 3 need to have to have the rate of duplication calculated and statistics added to claim differences.

3) In the epinephrine. norepinephrine experiments Fig 6 and lines 200-216 several issues need to be addressed. They use a lot of epinephrine 500uM is overkill. Moreover, were the plates protected from light at all times? Epinephrine oxidizes and is highly untable, the stocks have to be freshly prepared, used and tossed. All expriments need to be light protected. This information need to be provided. Moreover dopamine is NOT an inducer of QseC, if it acts in any way it is through another pathway.

4) Experiments regarding LEE expression should never be conducted on LB, where this island is poorly expressed and regulated in vitro. The standard in the field for both EHEC and EPEC is to express the LEE in DMEM low glucose. In fact KdpE does not regulate the LEE in high glucose. LB is fine to conduct flagella experiments in vitro.

**Part II – Major Issues: Key Experiments Required for Acceptance**

Reviewer #1: 1. Does the in vitro and in vivo QseB phosphorylation in the absence of QseC is PmrB-dependent?

2. Should be analyzed which motility genes (fliC, flhDC, among others) are affected by the transcriptional talk. Is there a consensus site for QseB-P in the motility genes? Does QseB-P bind to these promoter regions?

3. Should be corroborated if the QseB-binding site found for arn (and eptA) is recognized by QseB-P (site-directed mutagenesis). This analysis also applies for motility genes.

Reviewer #2: 1) The authors provide no complementation in their studies. They should consider complementing with wild-type QseB or a phosphomimetic to validate their results.

2) In EHEC and UPEC, QseB varies by a single amino acid and swapping the two does not alter QseB actvity. It would be good to provide the QseB and QseC sequence alignments between UPEC, EHEC and aEPEC.

3) One thought is that phylogeny may dictate QseBC regulation. Is the particular aEPEC strain used in E, A, or B2 clade?

4) Please provide the statistical test used for each analysis and in each figure. Just stating that GraphPad 9 was used is inadequate.

Reviewer #3: 1) either complement mutants or do whole genome sequencing to ensure that no secondary mutations are driving the phenotypes

2) Repeat LEE experiments in DMEM low glucose not LE, add some other genes besides ler (eae, espA, etc...) to qRT-PCR and do Western of EspB and/or EspA on secrete proteins. Several invetigators in the field can provide these antisera

3) calculate duplication time on growth curves

4) Fig 7A provide the actual luciferase data and add statistics

**Part III – Minor Issues: Editorial and Data Presentation Modifications**

Reviewer #1: (No Response)

Reviewer #2: (No Response)

Reviewer #3: the word data is plural, so data are and not data is should be used. There are several intances of this in the manuscript.

PLOS authors have the option to publish the peer review history of their article (what does this mean?). If published, this will include your full peer review and any attached files.

Reviewer #1: No

Reviewer #2: No

Reviewer #3: No
---

## [Decision Letter · Decision Letter 1]

6 Oct 2023

Dear Ms Fernandez-Ciruelos,

Thank you very much for submitting your manuscript "Cross-talk between QseBC and PmrAB two-component systems is crucial for regulation of motility and colistin resistance in Enteropathogenic *Escherichia coli*" for consideration at PLOS Pathogens. As with all papers reviewed by the journal, your manuscript was reviewed by members of the editorial board and by several independent reviewers. The reviewers appreciated the attention to an important topic. Based on the reviews, we are likely to accept this manuscript for publication, providing that you modify the manuscript according to the review recommendations.

I am returning your manuscript with two reviews. The reviewers came to the conclusion that the paper has been improved, as you will see.

There are, however, a few remaining minor revisions that need to be addressed to prepare the manuscript for publication.

Sincerely,

Thomas Guillard, PharmD, PhD

Academic Editor

PLOS Pathogens

David Skurnik

Section Editor

PLOS Pathogens

Kasturi Haldar

Editor-in-Chief

PLOS Pathogens

orcid.org/0000-0001-5065-158X

Michael Malim

Editor-in-Chief

PLOS Pathogens

orcid.org/0000-0002-7699-2064

I am returning your manuscript with two reviews. The reviewers came to the conclusion that the paper has been improved, as you will see.

There are, however, a few remaining minor revisions that need to be addressed to prepare the manuscript for publication.

Reviewer Comments (if any, and for reference):

Reviewer's Responses to Questions

**Part I - Summary**

Reviewer #1: The authors have addressed the main answers. They have demonstrated with genetic evidence the transcriptional cross-talk that shows the QseB phosphorylation by PmrB. In addition, the direct activation of QseB-P to arn promoter region was showed. However the authors should consider the following:

1. Which is the relevance of the QseB-binding site found on arn? It is known the transcription start site for arn? How could be the arn activation by QseB-P?

2. The authors analyze th expression of ler gene, which is the first gene of the LEE1 operon, one of the five operons that constitute the LEE island. The auhtors use these concepts interchangeably. Lines 242, 245, 253, 264, 259.

Reviewer #3: In their revised manuscript the authors addressed most of the comments and provided additional experimental data furthering their case for the mechanistic analysis of the cross-talk between the PmrAB and QseBC two-component systems in aEPEC.

The one comment that the authors failed to address is the lack of actual protein-protein interaction experiments. While the elegance of the site directed mutagenesis approach adopted in the study is highly commendable, the authors cannot reject the hypothesis of an indirect link between PmrAB and QseBC without actually showing phosphorylation of QseB by PmrB in their aEPEC strain.

In consequence, either these experiments should be performed or the phrasing of the manuscript should be modified to reflect the lack of direct observation of protein phosphorylation. For instance, L170 “PmrB is directly responsible for phosphorylation of QseB” or L306-308 “In the WT, QseB can be phosphorylated by the non-cognate PmrB HK, but it is immediately de-phosphorylated or sequestered by QseC (25,28)” are both very strong assertions when the experiment hasn’t been done by the authors in that aEPEC background.

Despite that one major shortcoming, the authors provide a very interesting addition to the field by underlining the diversity and complexity of histidine kinase-mediated regulation in E. coli.

Minor comments:

L201-202 “This data suggests that PmrAB TCS has an independent role in adapting to metal ion

stress that does not involve QseBC TCS” Since the authors are studying the effect of excess iron it might be interesting to note that a Fur-box has been identified in silico upstream of flhCD in the K12 strain. If this is also the case in the aEPEC strain studied, the lack of a functionalPmrAB system leading to increase of intracellular iron (according to reference 27) could also explain the decrease of motility observed.

L501-502 ‘Minimal inhibitory concentrations against EPEC O125ac:H6 were performed as described by European Committee on Antimicrobial Susceptibility Testing [54]’ → Is there a reason the authors chose to determine MIC in LB medium instead of the recommended Mueller Hinton broth?

Figure 4b and 5 → Isn’t the difference between WT and the ΔqseC mutant significant on both graphs?

Typos:

L42 ‘detect host signals as queues ‘ → cues

L380 ‘was provided obtained using Illumina-sequencing’

Figure 7B ‘lee expression’ → is it ler expression?

**Part II – Major Issues: Key Experiments Required for Acceptance**

Reviewer #1: The authors have addressed the main answers. They have demonstrated with genetic evidence the transcriptional cross-talk that shows the QseB phosphorylation by PmrB. In addition, the direct activation of QseB-P to arn promoter region was showed. However the authors should consider the following:

1. Which is the relevance of the QseB-binding site found on arn? It is known the transcription start site for arn? How could be the arn activation by QseB-P?

2. The authors analyze th expression of ler gene, which is the first gene of the LEE1 operon, one of the five operons that constitute the LEE island. The auhtors use these concepts interchangeably. Lines 242, 245, 253, 264, 259.

Reviewer #3: (No Response)

**Part III – Minor Issues: Editorial and Data Presentation Modifications**

Reviewer #1: The authors have addressed the main answers. They have demonstrated with genetic evidence the transcriptional cross-talk that shows the QseB phosphorylation by PmrB. In addition, the direct activation of QseB-P to arn promoter region was showed. However the authors should consider the following:

1. Which is the relevance of the QseB-binding site found on arn? It is known the transcription start site for arn? How could be the arn activation by QseB-P?

2. The authors analyze th expression of ler gene, which is the first gene of the LEE1 operon, one of the five operons that constitute the LEE island. The auhtors use these concepts interchangeably. Lines 242, 245, 253, 264, 259.

Reviewer #3: (No Response)

PLOS authors have the option to publish the peer review history of their article (what does this mean?). If published, this will include your full peer review and any attached files.

Reviewer #1: No

Reviewer #3: No

Figure Files:

Data Requirements:

Reproducibility:

References:

---

## [Editor Report · Decision Letter 2]

13 Nov 2023

Dear Ms Fernandez-Ciruelos,

We are pleased to inform you that your manuscript 'Cross-talk between QseBC and PmrAB two-component systems is crucial for regulation of motility and colistin resistance in Enteropathogenic *Escherichia coli*' has been provisionally accepted for publication in PLOS Pathogens.

Best regards,

Thomas Guillard, PharmD, PhD

Academic Editor

PLOS Pathogens

David Skurnik

Section Editor

PLOS Pathogens

Kasturi Haldar

Editor-in-Chief

PLOS Pathogens

orcid.org/0000-0001-5065-158X

Michael Malim

Editor-in-Chief

PLOS Pathogens

orcid.org/0000-0002-7699-2064
---

## [Editor Report · Acceptance letter]

30 Nov 2023

Dear Ms Fernandez-Ciruelos,

We are delighted to inform you that your manuscript, "Cross-talk between QseBC and PmrAB two-component systems is crucial for regulation of motility and colistin resistance in Enteropathogenic *Escherichia coli*," has been formally accepted for publication in PLOS Pathogens.

Best regards,

Michael Malim

Editor-in-Chief

PLOS Pathogens

orcid.org/0000-0002-7699-2064